# New-Generation Heterocyclic Bis-Pentamethinium Salts as Potential Cytostatic Drugs with Dual IL-6R and Mitochondria-Targeting Activity

**DOI:** 10.3390/pharmaceutics14081712

**Published:** 2022-08-17

**Authors:** Veronika Talianová, Zdeněk Kejík, Robert Kaplánek, Kateřina Veselá, Nikita Abramenko, Lukáš Lacina, Karolína Strnadová, Barbora Dvořánková, Pavel Martásek, Michal Masařík, Magdalena Houdová Megová, Petr Bušek, Jana Křížová, Lucie Zdražilová, Hana Hansíková, Erik Vlčák, Vlada Filimonenko, Aleksi Šedo, Karel Smetana, Milan Jakubek

**Affiliations:** 1BIOCEV, First Faculty of Medicine, Charles University, CZ-252 42 Vestec, Czech Republic; 2Department of Paediatrics and Inherited Metabolic Disorders, First Faculty of Medicine, Charles University and General University Hospital, Ke Karlovu 455/2, CZ-128 08 Prague, Czech Republic; 3Institute of Anatomy, First Faculty of Medicine, Charles University, CZ-120 00 Prague, Czech Republic; 4Department of Dermatovenerology, First Faculty of Medicine, Charles University and General University Hospital, CZ-128 08 Prague, Czech Republic; 5Department of Pathological Physiology and Department of Physiology, Faculty of Medicine, Masaryk University, Kamenice 5, CZ-625 00 Brno, Czech Republic; 6Institute of Biochemistry and Experimental Oncology, First Faculty of Medicine, Charles University, CZ-120 00 Prague, Czech Republic; 7Institute of Molecular Genetics, Academy of Sciences, CZ-140 00 Prague, Czech Republic

**Keywords:** IL-6R synthetic inhibitors, mitochondria, cancer

## Abstract

IL-6 signaling is involved in the pathogenesis of a number of serious diseases, including chronic inflammation and cancer. Targeting of IL-6 receptor (IL-6R) by small molecules is therefore an intensively studied strategy in cancer treatment. We describe the design, synthesis, and characteristics of two new bis-pentamethinium salts **5** and **6** (meta and para) bearing indole moieties. Molecular docking studies showed that both compounds have the potential to bind IL-6R (free energy of binding −9.5 and −8.1 kcal/mol). The interaction with IL-6R was confirmed using microscale thermophoresis analyses, which revealed that both compounds had strong affinity for the IL-6R (experimentally determined dissociation constants 26.5 ± 2.5 nM and 304 ± 27.6 nM, respectively). In addition, both compounds were cytotoxic for a broad spectrum of cancer cell lines in micromolar concentrations, most likely due to their accumulation in mitochondria and inhibition of mitochondrial respiration. In summary, the structure motif of bis-pentamethinium salts represents a promising starting point for the design of novel multitargeting compounds with the potential to inhibit IL-6 signaling and simultaneously target mitochondrial metabolism in cancer cells.

## 1. Introduction

Interleukin-6 (IL-6) is a multifunctional proinflammatory mediator that was originally identified as a T-cell-derived cytokine inducing terminal maturation in antibody-producing B cells [1]. It is known under different synonyms (interferon β2, 26 K factor, hybridoma growth factor, plasmocytoma growth factor, hepatocyte stimulatory factor, hematopoietic factor and cytotoxic T-cell differentiation factor) [2,3]. IL-6 signals through a receptor complex comprising IL-6Rα (CD126) and IL-6Rβ (CD130, glycoprotein 130 or gp130) and plays an important role in immune response, inflammation and hematopoiesis and in the endocrine and nervous systems [4,5].

The role of inflammation-supporting cytokines in cancer development and progression has recently become an active area in cancer research. As response rates in patients with advanced metastatic disease remain low with both standard chemotherapy and immunotherapy, novel approaches are being explored and cytokines have been proposed as potential therapeutic targets [6,7]. High levels of IL-6 in the tumor microenvironment, possibly reflecting the strong association between inflammation and cancer, have been found in almost all tumor types [3,8]. In addition, elevated serum concentrations of IL-6 correlate with worse prognosis in various cancers, including melanoma [9]. IL-6 plays a major role in the pathogenesis and development of malignancies, promotes tumor growth, inhibits apoptosis, induces angiogenesis and influences the overall metabolism in cancer patients [8,9,10]. In addition, IL-6 diminishes the effects of anticancer treatments by facilitating DNA repair and inducing antioxidant and anti-apoptotic effects in cancer cells. Therefore, blocking IL-6 or inhibiting its associated signaling alone or in combination with conventional anti-cancer therapies could be a potential therapeutic strategy in various cancers [10].

At present, targeting of IL-6 receptors (IL-6Rs) by monoclonal antibodies is an intensively studied therapeutic strategy [11,12,13,14]. Nevertheless, the inherent limitations of antibodies (e.g., high cost, invasive route of administration and high rate of immunogenicity) restrict their clinical usability [15]. The development of low-molecular-weight inhibitors is therefore highly desirable due to their superiority in in terms of oral absorption and low antigenicity. Despite the immense importance of this task and the efforts undertaken so far, only a few compounds are currently available [16,17,18,19,20,21,22,23,24,25,26].

Pentamethinium salts (PMSs) may represent an interesting structural motif for the design of new IL-6R inhibitors [3]. These compounds display mitochondrial accumulation [27,28,29,30] and potent cytotoxicity, which suggests their potential for use in cancer treatment [27,30,31,32]. For example, Krejci et al. reported that the cytotoxic effect of these compounds is associated with mitochondrial disintegration and metabolic collapse [30]. Thus, suitably designed PMSs could be used as dual anticancer agents targeting IL-6Rs and mitochondrial function simultaneously. This concept would effectively reflect the hypothesis that multitargeted therapeutic approaches capable of targeting/inhibiting more than one specific molecular target offer a promising strategy for anticancer treatment [33].

Nevertheless, the usability of PMSs can be significantly limited by their strong aggregation [27]. This could be overcome through a design based on imidazolium building blocks with higher charge density; e.g., PMS with two cationic charges. However, such a strategy may lead to a loss of cellular uptake and biological activity [28,31]. A possible solution could be Bis-PMSs, the charge density of which would lie in the interval between the values of mono- and doubly-charged PMS. Therefore, we designed, prepared and studied Bis-PMSs as potential cytostatic drugs with IL-6R inhibitory activity.

## 2. Materials and Methods

### 2.1. Synthesis of Bis-Pentamethinium Salts

All chemicals and solvents were purchased from commercial suppliers (Merck/Sigma-Aldrich; Prague, Czech Republic and TCI Europe; Eschborn, Germany) and used without further purification. Nuclear magnetic resonance (NMR) spectra were recorded with a 500 MHz instrument (Bruker BioSpin, Billerica, MA, USA) at room temperature in DMSO-*d*_6_ or CDCl_3_. The chemical shifts (δ) are presented in ppm, the coupling constants (J) in Hz. The MestReNova program v 14.2 (Advanced chemistry development, Toronto, ON, Canada) was used to process the NMR spectra. High-resolution mass spectrometry (HRMS) spectra were obtained using electrospray ionization (ESI) with an LTQ Orbitrap spectrometer. Flesh chromatography on silica was used for isolation of target compounds.

**Preparation of *p*-bis-vinamidinium salt 1 [34,35,36]:** Phosphorus oxychloride (7 mL; 75 mmol) was added dropwise to dry DMF (15 mL) at 0 °C. The reaction mixture was stirred at room temperature for 30 min. Then, a solution of *p*-phenylenediacetic acid (2 g; 10 mmol) in DMF (7 mL) was added and the reaction mixture was stirred at 80 °C overnight. The hot dark reaction mixture was then poured into crushed ice (ca. 200 mL), and a solution of sodium perchlorate (9.5 g; 78 mmol) in water (100 mL) was added to produce a precipitate. The solid was filtered, washed with water (3 × 75 mL) and vacuum-dried to obtain 5.01 g (95%) of a beige solid (1). ^1^H NMR (DMSO-*d*_6_): δ 2.46 (s, 12H), 3.25 (s, 12H), 7.40 (s, 4H), 7.73 (s, 4H) ppm. ^13^C NMR (DMSO-*d*_6_): δ 39.3, 48.7, 104.3, 132.2, 133.4, 163.0 ppm. HRMS (ESI^+^): calcd for C_20_H_32_N_4_ [M]^2+^ 164.13080; found 164.13102; calcd for C_30_H_32_Cl_2_N_4_O_4_ [M + ClO_4_]^+^ 427.21066; found 427.21078.

**Preparation of *m*-bis-vinamidinium salt 2 [37]:** Phosphorus oxychloride (7 mL; 75 mmol) was added dropwise to dry DMF (15 mL) at 0 °C. The reaction mixture was stirred at room temperature for 30 min. Then, a solution of *m*-phenylenediacetic acid (2 g; 10 mmol) in DMF (7 mL) was added and the reaction mixture was stirred at 80 °C overnight. The hot dark reaction mixture was then poured into crushed ice (ca. 200 mL), and a solution of sodium perchlorate (9.5 g; 78 mmol) in water (100 mL) was added to produce a precipitate. The solid was filtered, washed with water (3 × 75 mL) and vacuum-dried to obtain 4.59 g (87%) of a beige solid (2). ^1^H NMR (DMSO-*d*_6_): δ 2.50 (s, 12H), 3.24 (s, 12H), 7.26 (s, 1H); 7.40 (d, 2H, J = 7.7 Hz), 7.51 (*t*, 1H, J = 7.6 Hz) ppm. ^13^C NMR (DMSO-*d*_6_): δ 39.8, 48.5, 103.9, 128.5, 132.3, 133.4, 134.6, 163.3 ppm. HRMS (ESI^+^): calcd for C_20_H_32_N_4_ [M]^2+^ 164.13080; found 164.13105; calcd for C_30_H_32_Cl_2_N_4_O_4_ [M + ClO_4_]^+^ 427.21066; found 427.21082.

**Preparation of *p*-bis-malondialdehyde 3 [35,36]:** Vinamidinium salt 1 (1.47 g; 2.7 mmol) was added into an aqueous solution of sodium hydroxide (10% *w/w*; 10 mL) at 80 °C and the reaction mixture was stirred for 15 min (from suspension to yellow solution at the end). The reaction mixture was diluted with water (10 mL), cooled to ca. 5 °C and neutralized with 6 M hydrochloric acid to pH ≈ 5. Precipitated aldehyde was filtered, washed with water (20 mL) and vacuum-dried to obtain 518 mg (88%) of *p*-bis-malondialdehyde 3 as a white solid. ^1^H NMR (DMSO-*d*_6_): δ 7.38 (s, 4H), 7.50 (bs, 4H), 12.10 (bs, 2H) ppm. ^13^C NMR (DMSO-*d*_6_): δ 120.8, 128.3, 129.7, 188.0 ppm. HRMS (ESI^+^): calcd for C_12_H_11_O_4_ [M + H]^+^ 219.06573; found 219.06532; calcd for C_12_H_10_O_4_Na [M + Na]^+^ 241.04768; found 241.04737.

**Preparation of *m*-bis-malondialdehyde 4:** Vinamidinium salt 2 (1.47 g; 2.7 mmol) was added into an aqueous solution of sodium hydroxide (10% *w/w*; 10 mL) at 80 °C and the reaction mixture was stirred for 15 min (from suspension to yellow solution at the end). The reaction mixture was cooled to ca. 5 °C and neutralized with 1 M hydrochloric acid to pH ≈ 5. The mixture was left in the fridge overnight, and the precipitated aldehyde was then filtered, washed with water (20 mL) and vacuum-dried to obtain 301 mg (51%) of *m*-bis-malondialdehyde 4 as a beige solid. ^1^H NMR (DMSO-*d*_6_): δ 7.25 (m, 3H), 7.51 (s, 1H), 8.51 (bs, 4H), 10.25 (bs, 2H) ppm. ^13^C NMR (DMSO-*d*_6_): 121.1, 126.8, 127.5, 130.0, 130.8 ppm. HRMS (ESI^+^): calcd for C_12_H_11_O_4_ [M + H]^+^ 219.06573; found 219.06529; calcd for C_12_H_10_O_4_Na [M + Na]^+^ 241.04768; found 241.04738.

**Preparation of *p*-bis-pentamethinium salt 5 from *p*-bis-vinamidinium salt 1:** The compound was prepared using a slightly modified procedure described in the literature [34]. *p*-Bis-vinamidinium salt **1** (210 mg; 0.4 mmol), 1,2,3,3-tetramethyl-3H-indol-1-ium iodide (602 mg; 2 mmol) and potassium iodide (664 mg; 4 mmol) were mixed in anhydrous ethanol (6 mL). DIPEA (0.35 mL; 2 mmol) was added and the reaction mixture was stirred at 75 °C overnight. After cooling down, volatile compounds were removed under reduced pressure and the residue was purified using flash chromatography (dichloromethane/methanol, gradient from 98:2 to 90:0, *v/v*). *p*-Bis-pentamethinium salt **5** was obtained as a deep greenish-blue solid in a yield of 26 mg (6%).

**Preparation of *p*-bis-pentamethinium salt 5 from *p*-bis-malondialdehyde 3:** *p*-Bis-malondialdehyde **3** (44 mg, 0.2 mmol) and 1,2,3,3-tetramethyl-3H-indol-1-ium iodide (301 mg, 1 mmol) were mixed in *n*-butanol (15 mL). The reaction mixture was stirred at 115 °C overnight. After cooling down, volatile compounds were removed under reduced pressure and the residue was purified using flash chromatography (dichloromethane/methanol, gradient from 98:2 to 90:0, *v/v*). *p*-Bis-pentamethinium salt **5** was obtained as a deep greenish-blue solid in a yield of 79 mg (36%). ^1^H and ^13^C data were in accordance with the published data [34]. HRMS (ESI^+^): calcd for C_60_H_64_N_4_ [M]^2+^ 420.25600; found 420.25605; calcd for C_60_H_64_IN_4_ [M + I]^+^ 967.41702; found 967.41655.

**Preparation of *m*-bis-pentamethinium salt 6 from *m*-bis-vinamidinium salt 2:** The compound was prepared using a slightly modified procedure described in the literature [34]. *m*-Bis-vinamidinium salt 2 (210 mg; 0.4 mmol), 1,2,3,3-tetramethyl-3H-indol-1-ium iodide (602 mg; 2 mmol) and potassium iodide (664 mg; 4 mmol) were mixed in anhydrous ethanol (6 mL). DIPEA (0.35 mL; 2 mmol) was added and the reaction mixture was stirred at 75 °C overnight. After cooling down, volatile compounds were removed under reduced pressure and the residue was purified using flash chromatography (dichloromethane/methanol, gradient from 98:2 to 90:0, *v/v*). *m*-Bis-pentamethinium salt 6 was obtained as a deep greenish-blue solid in a yield of 22 mg (5%).

**Preparation of *m*-bis-pentamethinium salt 6 from *m*-bis-malondialdehyde 4:** Malondialdehyde 4 (44 mg, 0.2 mmol) and 1,2,3,3-tetramethyl-3H-indol-1-ium iodide (301 mg, 1 mmol) were mixed in *n*-butanol (15 mL). The reaction mixture was stirred at 115 °C overnight. After cooling down, volatile compounds were removed under reduced pressure and the residue was purified using flash chromatography (dichloromethane/methanol, gradient from 98:2 to 90:0, *v/v*). *m*-Bis-pentamethinium salt 6 was obtained as a deep greenish-blue solid in a yield of 59 mg (27%). ^1^H NMR (CDCl_3_): δ 1.78 (s, 24H), 3.56 (s, 12H), 5.82 (d, J = 14.0 Hz, 4H), 7.27 (m, 17H), 7.46 (d, J = 7.7 Hz, 2H), 7.82 (*t*, J = 7.7 Hz, 1H), 8.21 (d, J = 14.1 Hz, 4H) ppm. ^13^C NMR (CDCl_3_): δ 28.2, 33.2, 49.8, 101.0, 111.6, 122.2, 125.7, 129.0, 130.2, 130.8, 131.7, 133.7, 136.3, 140.8, 142.7, 153.1, 173.9 ppm. HRMS (ESI^+^): calcd for C_60_H_64_N_4_ [M]^2+^ 420.25600; found 420.25616; calcd for C_60_H_64_IN_4_ [M + I]^+^ 967.41702; found 967.41630.

### 2.2. Analytical Studies

Spectral characteristics in the range of 300 to 800 nm were obtained using a Shimadzu UV-2401 PC UV/VIS spectrometer. The Bis-PMSs were measured in four solvents (DMSO, methanol, ethanol and phosphate buffer). Both Bis-PMSs **5** and **6** were first dissolved in 1 mM DMSO stock solution, which was diluted to a final concentration of 1 μM, and measured in a classical 1 cm PMMA cuvette.

The fluorescence properties of the compounds were measured using a FLS1000 modular fluorescence spectrometer (Edinburgh Instruments, Livingston, UK). The substances were dissolved in DMSO and diluted to a final concentration of 500 nM. The measurements were carried out in a Fluorescence Quartz Cuvette Cell with 1 cm length for the absorbent layer. The excitation and emission maxima of Bis-PMSs **5** and **6** are provided in Appendix A.

### 2.3. Photostability Assay

The photostability of the fluorescent probes was tested in the BJ-hTERT cell line. The cells (10,000 cells per well) were seeded in 35 mm glass-bottom dishes in complete culture media and allowed to adhere for 24 h. When the cells had reached the desired confluence, the medium was removed and the cells were washed twice with PBS and incubated in Dulbecco’s Modified Eagle‘s Medium (DMEM), lacking phenol red and containing bis-pentamethinium salts **5** and **6** (at 100 nM) and MT-G (100 nM) or MT-R (100 nM), for 30 min under standard conditions.

The excitation wavelength was 630 nm and the emission was recorded in the 650–670 nm range. For MT-G, the excitation wavelength was 490 nm and the emission was measured in the 510–550 nm range. MT-R was excited at 581 nm and the emission was recorded in the 640–650 nm range. Exposure time was 5 min.

### 2.4. In Silico Docking of Bis-Pentamethinium Salts 5 and 6 to the IL-6R Model

The 3D structural model of the human interleukin 6 receptor (IL-6R) was generated from the crystallographic structure of the complex between IL-6 and its receptor (IL-6R-α and IL-6R-β, PDB id 1P9M) by editing out the coordinates of the IL-6 molecule. All bound water molecules and ligands were removed.

Bis-PMSs **5** and **6** and a mono penthamethinium salt with a phenyl group in the γ-position were docked to the IL-6R using the CB-Dock web server [38]. All docking poses from this cavity (binding site) were visually inspected with BIOVIA Discovery Studio Visualiser [39], and 2D interaction diagrams and 3D visualizations of the protein–ligand complex were generated and are presented.

### 2.5. Microscale Thermophoresis (MST)

Microscale thermophoresis (MST) measurements were performed with a Monolith NT.115 device (Nano Temper Technologies GbmH, installed at the Centre of Molecular Structure, BIOCEV, with access granted through CIISB). The binding affinities of Bis-PMSs **5** and **6** to the IL-6R receptor molecule (Bio-Techne R&D Systems, Minneapolis, MN, USA; containing α and β subunits) were measured in PBS buffer (pH 7.4). Sixteen-step dilution was performed by adding 20 µL of IL-6R at twice the concentration (10 µmol/L) to the first PCR tube, and 10 µL of buffer (PBS, pH = 7.4) was added to PCR tubes 2–16. Then, we transferred 10 µL of the ligand from PCR tube 1 to PCR tube 2 and mixed the content by pipetting up and down multiple times. We repeated this procedure for tubes 3–16 and discarded the last 10 µL from PCR tube 16. The first capillary contained the highest concentration of the ligand (e.g., IL-6R) and the last capillary (number 16) contained the lowest concentration of the ligand. Ten µL of each Bis-PMS (**5** or **6)** in PBS buffer with 1% of DMSO and 0.005% of Pluronic was added to each PCR tube and mixed by pipetting up and down. The final concentration of Bis-PMS (**5** and **6**) was 100 nmol/L. We loaded solutions from each PCR tube into NT.115 Standard Capillaries, and scanning was performed using 20% LED power and 40% MST power. Data were analyzed using Nano Temper Technologies’ NT Analysis 1.5.41 software (Technologies GmbH; München, Deutschland).

### 2.6. Cell Lines and Cell Culture

Cells were cultured under standard conditions in a humidified atmosphere of 5% CO_2_ in air at 37 °C. Normal dermal fibroblasts (HF-P4) were derived from residuary skin specimens of healthy persons who underwent plastic surgery at the Department of Plastic surgery, Third Faculty of Medicine, Charles University and University Hospital Královské Vinohrady, Prague, Czech Republic, as described by Szabo et al., 2011 and Dvořánková et al., 2019 [9,10]. Informed consent from the patients and the approval of the Local Ethical Committee, in agreement with the Helsinki Declaration, were obtained. hTERT-immortalized foreskin fibroblast cell line **BJ-hTERT** (ATCC CRL-4001) was cultured in Dulbecco’s Modified Eagle’s Low-Glucose Medium, supplemented with 10% FBS, 0.01% non-essential amino acids (NEAAs), 100 U/mL penicillin and 100 µg/mL streptomycin. Human skin melanoma cell line **A-2058**, derived from a lymph node metastasis (ATCC CRL-11147, Manassas, VA, USA), and the highly metastatic **BLM** melanoma cell line (obtained from L. van Kempen and H. Van Krieken, Department of Pathology, Radboud University, Nijmegen, The Netherlands) were cultured in Dulbecco’s Modified Eagle’s Medium (DMEM) supplemented with 10% FBS, 100 U/mL penicillin and 100 µg/mL streptomycin. The osteosarcoma **U-2 OS** cell line (ATCC HTB-96) was cultured in McCoy’s 5a modified medium supplemented with 10% FBS, 100 U/mL penicillin and 100 µg/mL streptomycin. The non-small cell lung human carcinoma **H1299** cell line (ATCC CRL-5803) was cultured in RPMI-1640 medium supplemented with 10% FBS, 100 U/mL penicillin and 100 µg/mL streptomycin. Human breast tumor cell line **BT-20** (ATCC HTB-19) was cultured in Eagle’s Minimum Essential Medium (EMEM) supplemented with 10% FBS, 100 U/mL penicillin and 100 µg/mL streptomycin. Human (**U251MG**, Cell Lines Service GmbH, Eppelheim, Germany) and mouse (**Gl261**, Division of Cancer Treatment and Diagnosis Tumor repository, National Cancer Institute, Bethesda, MD, USA) glioblastoma cell lines were propagated in Dulbecco’s Modified Eagle’s Medium (DMEM) and RPMI-1640, respectively, supplemented with 10% of fetal bovine serum (FBS).

### 2.7. Cytotoxicity Assays

A colorimetric MTT cell metabolic activity assay was used to determine the cytotoxicity of Bis-PMSs **5** and **6**. For these experiments, the HF-P4, BJ-hTERT, BLM, A-2058, U-2 OS, H1299 and BT-20 cells were cultured under standard conditions (37 °C, humidified atmosphere with 5% CO_2_ in air) in culture medium (DMEM, RPMI-1640, McCoy’s or EMEM with 10% FBS). Cells were plated in 96-well plates at a density of 5000 cells per well and grown for 24 h. The medium was then replaced with a medium containing Bis-PMSs **5** and **6**, and cells were cultured for an additional 48 h. Afterwards, the medium was exchanged for medium with the MTT yellow tetrazolium dye (3-(4,5-dimethylthiazol-2-yl)-2,5-diphenyltetrazolium bromide, Sigma-Aldrich) and incubated for 2 h at 37 °C. After that, the yellow tetrazole dye was removed and DMSO was used to dissolve the reduced purple formazan formed in living cells. The absorbance of the converted dye was measured at a wavelength of 570 nm with the reference of 630 nm using a SpectraMax ABS Plus microplate reader (Molecular Devices). Experiments were performed in quadruplicate and repeated three times with similar results. In U251MG and GL261 cells, an assay assessing cell growth independent of mitochondrial respiration was used [40]. Cells were plated in 96-well plates at a density of 5000 cells per well and grown for 24 h under standard conditions in culture medium (DMEM and RPMI-1640 with 10% FBS for U251MG and Gl261 cells, respectively). The medium was then replaced by that supplemented with Bis-PMSs **5** and **6**. After 72 h of exposure, surviving cells were fixed and stained with methylene blue (5 g/L in 50%, *v/v*, ethanol) and lysed with 1% sodium dodecyl sulfate, and the relative cell number was determined by reading absorbance at 630 nm using a 96-well plate reader (Sunrise; Tecan, Männedorf, Switzerland).

The cell inhibitory concentration (IC) was calculated using the equation IC = (A_PMS_ well/mean A_control_ wells) × 100%. The half-maximal inhibitory concentration (IC_50_) was calculated from dose–response curves using GraphPad Prism software (v. 8.0.1, La Jolla, CA, USA). The selectivity indexes towards cancer cells were calculated as a ratio of the average of the IC_50_ values for HF-P4 and BJ-hTERT human fibroblasts and the IC_50_ value for the corresponding cancer cell line. For U251MG and Gl261 glioblastoma cells, the cytoselectivity was calculated against normal HF P4 fibroblast cells, which were analyzed using the same cytotoxicity assay.

### 2.8. Intracellular Studies of Bis-Pentamethinium Salts

Intracellular localization of Bis-PMSs **5** and **6** in living cells was investigated at 37 °C and 0.05 CO_2_ tension using a Leica TCS SP8 WLL SMD-FLIM confocal microscope equipped with an HC PL APO CS2 63x/1.2 W water immersion objective. Glass bottom dishes (dish size 35 mm, well size 20 mm), especially designed for high-resolution imaging, were used. Human dermal fibroblast cells (HF-P4—primary cell culture or BJ-hTERT cell line), melanoma cells (A-2058, BLM), osteosarcoma cells (U-2 OS), non-small cell lung cancer cells (H1299) and breast cancer cells (BT-20) were applied to the glass bottom dishes at a density of 10,000 cells per well. All cells were kept under standard culture conditions in culture medium (DMEM, RPMI-1640, McCoy’s or EMEM with 10% FBS). When the cells reached the desired confluence (after 24 h), the medium was removed from the dish and the cells were rinsed twice with PBS and further incubated for 30 min in complete culture medium supplemented with the Bis-PMS at a concentration of 100 nM. A commercially available organic fluorescent probe, MitoTracker™ Green FM (ThermoFisher Scientific, Waltham, MA, USA), was used to label mitochondria. This green fluorescent dye was added at 300 nM concentration to the cells, which were then incubated for 30 min under standard growth conditions. After removal from the incubator, the cells were washed twice with PBS and analyzed. For the measurements, the excitation wavelength was 630 nm and the emission was measured in the 650–670 nm range. The MitoTracker™ Green FM dye was excited at 490 nm and the emission was recorded in the 510–550 nm wavelength range.

For electron microscopy, control HFP4 and melanoma A2058 cells, as well as cells exposed to 5 µM Bis-PMS **5** or **6** for 2 h, were routinely processed. Briefly, cells were fixed with 2.5% glutaraldehyde in 0.1 M Sörensen´s sodium-potassium phosphate buffer at pH 7.2–7.4 (all Thermo Fisher Scientific, Waltham, MA, USA). After washing, the cells were postfixed in 1% osmium tetroxide, dehydrated with a series of acetone (Lach-Ner, Neratovice, Czech Republic) and embedded in Epon-Durcupan resin (Sigma-Aldrich, St. Louis, MO, USA). Then, 80 nm ultrathin sections were prepared using an Ultramicrotome Leica EM UC6 (Leica Microsystems, Wetzlar, Germany) with a diamond knife (Diatome, Biel, Switzerland). The sections were mounted on 200 mesh size copper grids and examined in a JEOL JEM-1400 Flash transmission electron microscope operated at 80 kV equipped with a Matataki Flash sCMOS camera (JEOL, Akishima, Tokyo, Japan).

### 2.9. Colocalization Analysis

Colocalization analysis was performed in Fiji-Image J software using images of cells exposed to Bis-PMSs **5** and **6** and MitoTracker™ Green FM. The correlation between fluorescence intensities in the green and red channels was evaluated using Pearson’s correlation coefficient.

### 2.10. Mitochondrial Respiration

Mitochondrial respiration and extracellular acidification rate were measured in an Agilent Seahorse XF Analyzer (XF24, Agilent, Prague, Czech Republic) following Zdrazilova et al. [41], with two slight modifications. First, bis-pentamethinium salts **5** and **6** were added to cells in concentrations of 1 and 5 µM each—always into four wells on the day after seeding—and were incubated for two hours. Second, 50 µL of glucose (final assay concentration 10 mM) was added to Cartridge port A, 55 µL oligomycin (final assay concentration 1.5 µM) to port B, 61 µL FCCP (final assay concentration 0.6 µM) to port C and 67 µL rotenone with antimycin A and 2-deoxy-D-glucose (final assay concentrations 2 uM, 1 µM and 100 mM) to port D.

### 2.11. Inhibitory Effect of Bis-PMSs 5 a 6 on STAT3 Activation in a HEK Reporter Cell Line

Reporter cells (HEK-Blue™ IL-6 Cells (InvivoGen, San Diego, CA, USA); 10,000 cells per well in a 96-well plate) were seeded in DMEM with 10% heat-activated FBS (56  °C for 30 min to inactivate the alkaline phosphatase activity) and allowed to fully adhere for 48 h.

After 48 h, the medium was changed. All experiments were performed in three technical replicates. The tested concentrations of Bis-PMSs **5** and **6** in the culture medium were 10, 5, 1 and 0.5 µM. After 2 h (time to inhibit the receptor complex), recombinant human IL-6 (10 ng/mL) was added, and cells were further cultured in an incubator at 37 °C and 5% CO_2_ for 24 h. Tocilizumab (20 ng/mL) in the culture medium was used as a control inhibitor under otherwise identical conditions.

After 24 h of incubation with IL-6, a 20 µL sample was taken from each well following a brief agitation of the culture plate (on a shaker). The sample was mixed with 180 µL of QuantiBlue substrate to quantity the activity of secreted embryonic alkaline phosphatase (SEAP). The optical density (absorbance) of the color product was measured using a microplate reader (SpectrMax iD3 spectrophotometer at 635 nm) after 30 min at 37 °C.

## 3. Results and Discussion

### 3.1. Synthesis of Bis-Pentamethinium Salts

Generally, pentamethinium salts can be prepared by reaction of quaternary salts of nitrogen heterocycles containing an active methyl (e.g., *N*-alkylated 2,3,3-trimethyl-3H-indol-1-ium, 2-methylbenzothiazol-3-ium, 2-methylbenzoxazol-3-ium, 2- or 4-methylquinolin-1-ium salts) with malondialdehydes or with vinamidinium salts [28,29,31,34,42,43,44].

The synthesis of Bis-PMS is shown in Figure 1. The initial *p*- and *m*-phenylenediacetic acid was reacted with phosphorus oxychloride in DMF to yield the bis-vinamidinium salts **1** and **2** [34,35,36]. Bis-vinamidinium salts were isolated as perchlorates in 87–95% yields. In the next step, bis-vinamidinium salts **1** and **2** were hydrolyzed by aqueous sodium hydroxide to give corresponding bis-malondialdehydes in 51–88% yields [36]. In the last step, bis-malondialdehydes **3** and **4** were reacted with 1,2,3,3-tetramethyl-3H-indol-1-ium iodide to give target Bis-PMSs **5** and **6** in 27–36% yields. Alternatively, target Bis-PMSs **5** and **6** were prepared by reaction of vinamidinium salts **1** and **2** with 1,2,3,3-tetramethyl-3H-indol-1-ium iodide in 5–6% yields. [36] In all cases, reactions with *m*-derivatives gave products (compounds **2**, **4,** and **6**) in lower yields compared to *p*-derivatives.

### 3.2. Photophysical Characteristics

The spectral characteristics of Bis-PMSs **5** and **6** were obtained using UV/VIS and fluorescent spectrometric studies. Both salts were measured in four different solvents: dimethyl sulfoxide (DMSO), ethanol, methanol and phosphate-buffered saline. Absorption and fluorescence spectra are shown in Figure 2 and Appendix A and the resulting values for the excitation and emission maxima are provided in Appendix A.

Both Bis-PMSs had absorption/excitation maxima in the purple area of the visible spectrum (600–640 nm) and emission maxima in the red area (650 to 670 nm). The meta derivative (Bis-PMS **6**) showed higher fluorescence intensity at the wavelength of the emission maxima compared to the para derivative, Bis-PMS **5** (Figure 2). These characteristics were similar to mono-substituted indolium pentamethinium salts. For example, a DMSO solution of a pentamethinium salt with pyrimidine substitution in the γ-position displayed an absorption maximum at 643 nm, an emission maximum at 655 nm and an excitation maximum at 641 nm [28].

Stability of Bis-PMS **5** and PMS **6** in cell cultivation medium (EMEM) simulating physiological conditions was evaluated using UV-Vis spectroscopy. Both compounds, especially Bis-PMS 5, displayed significant spectral change over time (Appendix A), which could imply their possible chemical instability in aqueous solutions, most probably their hydrolysis. Nevertheless, this effect was strongly repressed in the presence of MeOH (another protic solvent), which demonstrated that the observed spectral change was not caused by hydrolysis but most probably by interaction with media components, such as anions, and/or aggregation of the compounds. UV-Vis spectra in methanol and phosphate buffer, which represent monomer and aggregate forms of polymethinium salts, are shown in Appendix A. Similar spectral behavior was previously observed for mono pentamethinium salts [27,28,29,31,45,46,47].

The photostability of Bis-PMSs **5** and **6** was similar to MitoTracker™ Green FM and significantly better than MitoTracker™ Red FM (Appendix A).

### 3.3. In Silico Docking of Bis-Pentamethinium Salts **5** and **6** to the IL-6R Model

For the docking studies, we used a structural model of IL-6R consisting of IL-6Rα and IL-6Rβ (gp130) subunits without IL-6 (pid 1P9M). Bis-PMSs 5 and 6 were docked to the IL-6R using the CB-Dock web server [38]. All docking poses from the protein binding site were visually inspected with BIOVIA Discovery Studio Visualiser [39], and 2D interaction diagrams and 3D visualizations of the protein–ligand complex were generated and are presented (Figure 3). The binding model suggests that both Bis-PMSs **5** and **6** bind to the IL-6Rβ subunit responsible for signal transduction after IL-6 binding [48]. A similar mechanism was observed for other low-molecular-weight inhibitors of IL-6 signaling, such as madindoline [24]. The computed values of the free energy of binding between Bis-PMSs **5** and **6** and IL-6R were −9.5 and −8.1 kcal/mol, respectively, which suggests a strong affinity in both compounds for the IL-6R and, thus, possible inhibitory activity towards IL-6R signaling.

Docking studies suggest that both compounds bind to IL-6R via their interaction with Phe69, Ile52, Ile77, Ala73 and Phe50. Nevertheless, we found a significant difference in the mode of interaction (Appendix A). For example, Bis-PMS **5** forms an alkyl bond with Ile52 and Ile77 through its methyl group. The methyl group of Bis-PMS **6** forms an alkyl bond with Ile77; however, IL77 also interacts via alkyl-π bond with the methinium chain. Ile52 forms a π-alkyl bond with the benzene imidazolium group. Bis-PMS **5** interacts with Phe69 (π-aliphatic bond with methyl groups) and Phe50 (π-bond with the benzene imidazolium group and π-alkyl bond with the methyl group). In the case of Bis-PMS **6**, Phe50 interacts with two imidazolium benzene groups via π-sigma and π-alkyl with the methyl group and methinium chains, respectively. Similarly, the phenyl group of Phe50 π stacks and forms a π-alkyl bond with the imidazolium benzene and methyl groups, respectively.

Furthermore, only the methyl group of Bis-PMS **5** interacts with Asn48 (alkyl bond) and His49 (π donor hydrogen bond). On the other hand, Bis-PMS **6** exclusively interacts with Pro53 (π-alkyl with the benzene imidazolium group). This suggests that the para structure motif of **5** could be more suitable for the binding of IL-6R. In this case, its shift to the meta-structure motif represented by Bis-PMS **6** leads to significantly reduced binding (binding energy −9.5 vs. −8.1 kcal/mol). For comparison, the value of binding energy obtained from docking of a mono pentamethinium salt with a phenyl group in the γ-position (Appendix A) was −8.3. kcal/ mol.

### 3.4. Interaction of Bis-Pentamethinium Salts 5 and 6 with IL-6R

The ability of Bis-PMS **5** and **6** to bind to the IL-6R protein containing α and β subunits was assessed by microscale thermophoresis (Figure 4). We processed the temperature jump (T-jump) defined as the fluorescence change induced by sample heating before the thermophoretic molecule transport sets in. T-jump allows the quantification of the affinity of interaction from changes in the intrinsic temperature dependence of the fluorescence.

The experimentally determined dissociation constants for Bis-PMSs **5** and **6** were 26.5 ± 2.5 nM and 304 ± 27.6 nM, respectively, which agreed with the docking results (see the in silico data above). In accordance with our expectations (based on the value of interaction energy), higher affinity for the IL-6R was observed for Bis-PMS **5**. Nevertheless, the values in both compounds suggest their strong potential for IL-6R inhibition. For comparison, tocilizumab (the antibody used in clinical practice) binds Il-6R with a significantly higher affinity (Kd = 2.54 ± 0.12 nM) [49], and values previously reported for synthetic inhibitors, such as chikusetsusaponin inhibitors, are several times higher compared to Bis-PMS **5** and **6** [50]. In line with this, we observed a strong inhibitory effect of tocilizumab on IL-6-induced STAT3 activation in IL-6 reporter cells and a mild effect of nontoxic concentrations of Bis-PMSs **5** and **6** (Appendix A). Thus, our data imply that the Bis-PMS scaffold is a promising novel structure motif for the inhibition of IL-6R signaling.

### 3.5. Cytotoxicity Assays

The cytotoxic activities of Bis-PMSs **5** and **6** were evaluated in normal (HF-P4 and BJ-hTERT fibroblasts) and malignant (BLM, A-2058, U-2 OS, H1299, BT-20) cells using a conventional MTT viability assay. For the measurements, cells were exposed to 0–10 µM of Bis-PMSs **5** and **6** under standard conditions in a complete cell culture medium and cell viability was measured after 48 h of treatment. In addition, cytotoxicity was tested in U251MG and Gl261 glioma cells using an assay that assesses cell growth independent of mitochondrial respiration [40] (Appendix A). The mean IC_50_ values are shown in Table 1 and Appendix A. The cytoselectivity index for cancer cell lines is shown in Figure 5.

Bis-PMS **6** exhibited higher toxicity in most tested cell lines (Table 1). The cellular viability of cancer cells was 80% and less after 48 h of incubation with the compound at >0.5 μM in the majority of cell lines. For comparison, the IC_50_ of mono pentamethinium salts evaluated in previous studies was comparable [28]. These results suggest the cytotoxicity of the tested compounds against a broad spectrum of cancer cells representing various tumor types. Nevertheless, there was substantial toxicity towards normal cells as well.

### 3.6. Influence of Bis-Pentamethinium Salts 5 and 6 on Mitochondrial Morphology and Function

PMS as hydrophobic cations are usually localized in mitochondria [27,28,29]. They can display strong cytotoxic effects against cancer cells closely associated with alterations of the mitochondrial metabolism [30].

Localization of the Bis-PMSs **5** and **6** was studied in normal fibroblasts (a primary cell culture HF-P4 and a cell line BJ-hTERT), two melanoma cell lines (BLM and A-2058), an osteosarcoma cell line (U-2 OS) and a non-small cell lung cancer cell line (H1299). No obvious cell morphology or viability changes were observed with the used concentration (100 nM). Both Bis-PMSs **5** and **6** displayed a nearly exclusive mitochondrial localization (Figure 6, Appendix A), as demonstrated by an almost complete overlap of their fluorescence signal and the fluorescence signal of the specific commercial mitochondrial probe MitoTracker Green (MT-G) (correlation coefficients (r) between 0.94 and 0.98; Appendix A).

As both Bis-PMSs **5** and **6** rapidly accumulated in the mitochondria in all cell types used in this study, we investigated their possible effect on mitochondrial respiration by employing the Seahorse Bioanalyzer. Parameters reflecting mitochondrial function, such as routine respiration, leak respiration, electron transfer capacity and ATP production, were measured in melanoma A2058 (Figure 7), glioblastoma U251MG and fibroblast HFP4 cells (Appendix A).

While 1 μM Bis-PMS **5** had no effect on mitochondrial respiration, 5 μM Bis-PMS **5** and both concentrations of Bis-PMS **6** caused a significant reduction in basal and maximal respiration in A2058 cells. These effects were accompanied by enhanced glycolysis as determined by extracellular acidification rate (ECAR) (Figure 7, Appendix A).

We further evaluated the effect of Bis-PMS on mitochondria using electron microscopy and observed significant changes in mitochondrial morphology in normal fibroblasts, as well as in melanoma cells exposed to Bis-PMS at a concentration inducing metabolic changes (5 µmol/L). The morphological changes were particularly pronounced for Bis-PMS **5** (Figure 6).

After application of Bis-PMSs **5** and **6**, the majority of mitochondria were swollen with only marginal cristae or cristolysis, a clear matrix with a web-like structure and dense aggregates in the lumen (Figure 6, Appendix A). The mitochondria in control cells without the application of the compounds were without pathological changes and showed typical morphology, a dense matrix and a well-developed system of cristae (Appendix A). Similar morphological changes were induced by Bis-PMSs **5** and **6** in HFP4 fibroblast cells, but they were somewhat smaller than in A2508 melanoma cells (Figure 6, Appendix A).

Collectively, our results suggest that both Bis-PMSs were cytotoxic for the tested cells, most probably due to their effect on mitochondria, but participation of IL-6R targeting is also possible. Similar results were described for other types of PMSs [27,29,31,32]. Their cytotoxic effects were also caused by the targeting of mitochondria, as summarized by Leischner Fialova [51]. In the present study, we clearly showed metabolic differences, such as reduced mitochondrial respiration and ensuing increased glycolysis as a compensatory mechanism, in cells treated with Bis-PMSs **5** and **6**. In addition, data published in this study demonstrate that both Bis-PMSs **5** and **6** interact with the receptor for IL-6. IL-6 is produced by cancer-associated fibroblasts and cancer cells and is an important molecule for the exchange of information in the cancer ecosystem [7]. Thus, a dual cytotoxic effect in both Bis-PMSs may be expected.

Does dual targeting make sense? There is a relationship between gp130 and respiration. It has been found that STAT3 participates in the control of cellular respiration in the mitochondria (e.g., complexes I and II of the electron transport chain) [52,53]. In addition, activated STAT3 in the mitochondria can strongly support tumorigenesis [54,55]. In line with this, repression of Il-6-induced STAT3 mitochondrial localization caused increased levels of mitochondrial ROS, collapse of the mitochondrial membrane potential and, subsequently, apoptosis of cancer cells [45].

It should be noted in this context that inhibition of IL-6R and/or gp130 alone can repress cell proliferation, but it is not associated with the activation of apoptotic pathways [3]. Nevertheless, Bis-PMSs **5** and **6** and mono pentamethinium salts also display significant cytotoxicity [28,29]. Due to their fast mitochondrial localization, this effect is most probably associated with alteration of mitochondrial metabolism and/or functionality [29]. Based on these data, the possible mechanism of the anticancer effects of both tested compounds may include inhibition of gp130 signaling, resulting in decreased cell proliferation, and most probably also sensitization of mitochondria to their direct targeting (Figure 8).

This dual alteration of mitochondrial metabolism (inhibition of gp130 and direct mitochondrial targeting) could be an effective method for inducing apoptosis in cancer cells with activated IL-6 signaling. However, other studies are needed to optimize, or reject, this novel therapeutic strategy.

Our results suggest that the structural motif of bis-pentamethinium salts could be applied in the design of novel anticancer drugs for the combined targeting of IL-6R and mitochondrial respiration. There are nevertheless several challenging aspects.

Firstly, simultaneous inhibition of mitochondrial respiration and IL-6R can be complicated by the different subcellular localization of the targets. IL-6R is localized on the cell surface, while targeting mitochondrial respiration requires efficient mitochondrial uptake. Pentamethinium salts with aromatic substitution in the γ-position usually display rapid mitochondrial localization due to strong affinity for cardiolipin [28,29]. For example, mitochondrial localization of indolium pentamethine can be observed after 5–10 min [28]. In the case of Bis-PMSs **5** and **6**, mitochondrial uptake was not that fast. This intracellular accumulation may nevertheless limit the availability of the compounds to inhibit IL-6R. On the other hand, pentamethinium with aromatic substitution in the γ-position could represent a promising structure motif for the targeting of pathological mitochondria, such as those with lower mitochondrial potential [28,29]. It is well-known that cancer cells can display significantly decreased mitochondrial membrane potential compared to healthy cells [56].

Secondly, although gp130 (IL-6Rβ) is an essential part of the IL-6R and its inhibition blocks signal transduction, it also participates in the signaling of other members of the IL-6/IL-12 family (e.g., IL-11, IL-23 and IL-35) [57]. The binding of both bis-pentamethinium salts to gp130 thus implies they not only block IL-6 signaling. However, in terms of their possible therapeutic application, this may be beneficial. For example, IL-11 activates IL-11RA and gp130/STAT3 signaling and plays an important role in tumorigenesis. Its higher levels can stimulate tumor immune evasion and metastatic spread and are associated with disease progression and poor patient outcome [58].

Thirdly, although both Bis-PMSs **5** and **6** decreased secretion of alkaline phosphatase controlled by STAT3 in the HEK reported cells, the inhibitory effect was substantially lower compared to tocilizumab (Appendix A). Furthermore, we cannot exclude the possibility that the decreased alkaline phosphatase secretion could have been partially caused by the effect of the compounds on mitochondria.

Fourthly, selectivity for cancer cells remains an important issue. Although bis-PMSs **5** and **6** represent promising compounds, they also inhibit growth and metabolic activity in non-malignant cells. Further studies are needed to identify derivatives with selective accumulation in the tumor tissue and/or more selective effects on cancer cells.

## 4. Conclusions

New-generation heterocyclic bis-pentamethinium salts **5** and **6** were prepared from phenylenediacetic acids, which were converted into corresponding bis-vinamidinium intermediates (**1, 2**) and, subsequently, to bis-malondialdehydes (**3**, **4**). The reaction of quaternary indolium salt with bis-vinamidinium salts (**1**, **2**) or with bis-malondialdehydes (**3**, **4**) produced target the Bis-PMSs (**5**, **6**) in low and moderate yields, respectively.

Docking studies and microscale thermophoresis assays showed that both bis-pentamethinium salts **5** and **6** displayed high affinity for IL-6R. In addition, both compounds rapidly accumulated in mitochondria, repressed mitochondrial respiration with compensatory enhancement of glycolysis and altered mitochondrial morphology. These effects are most likely responsible for the cytotoxicity of the compounds towards a wide spectrum of cancer cells. The new Bis-PMSs thus represent a promising structure motif for multitargeting compounds interfering with IL-6 signaling and simultaneously impacting mitochondria, an emerging target for anticancer therapy [59]. Future studies modifying their structure to increase the selectivity for cancer cells are warranted.

## Figures and Tables

**Figure 1 pharmaceutics-14-01712-f001:**
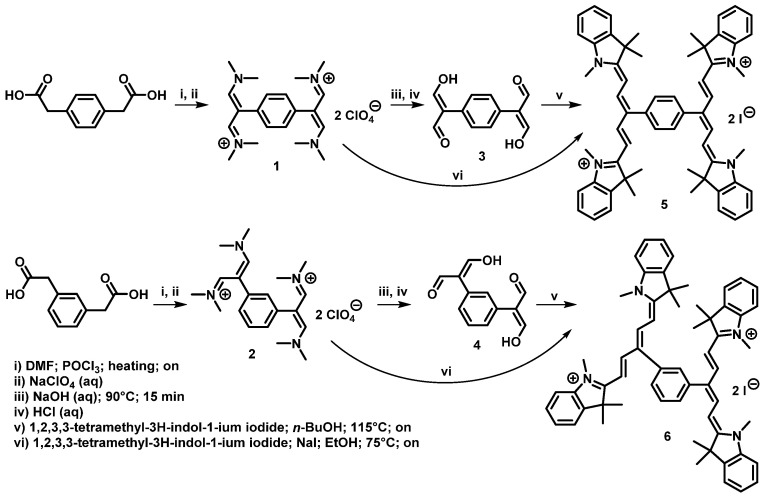
Synthesis of Bis-PMSs **5** and **6** overnight (on).

**Figure 2 pharmaceutics-14-01712-f002:**
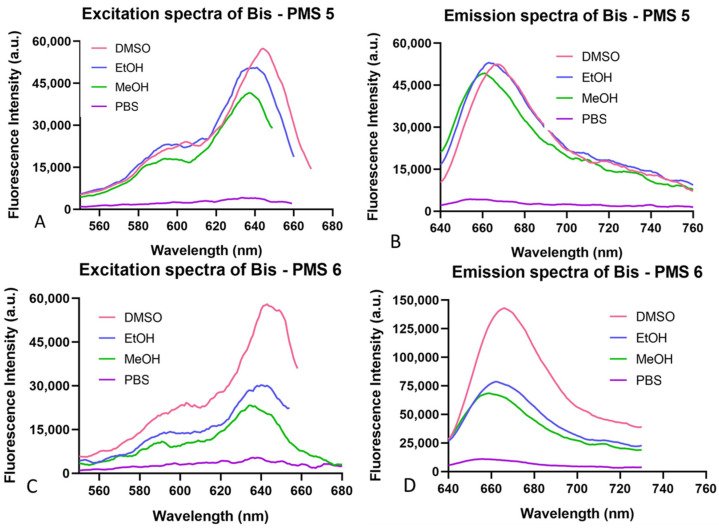
Excitation (**A**,**C**) and emission (**B**,**D**) spectra of Bis-PMSs **5** and **6** in four different solvents (DMSO, EtOH, MeOH and phosphate-buffered saline).

**Figure 3 pharmaceutics-14-01712-f003:**
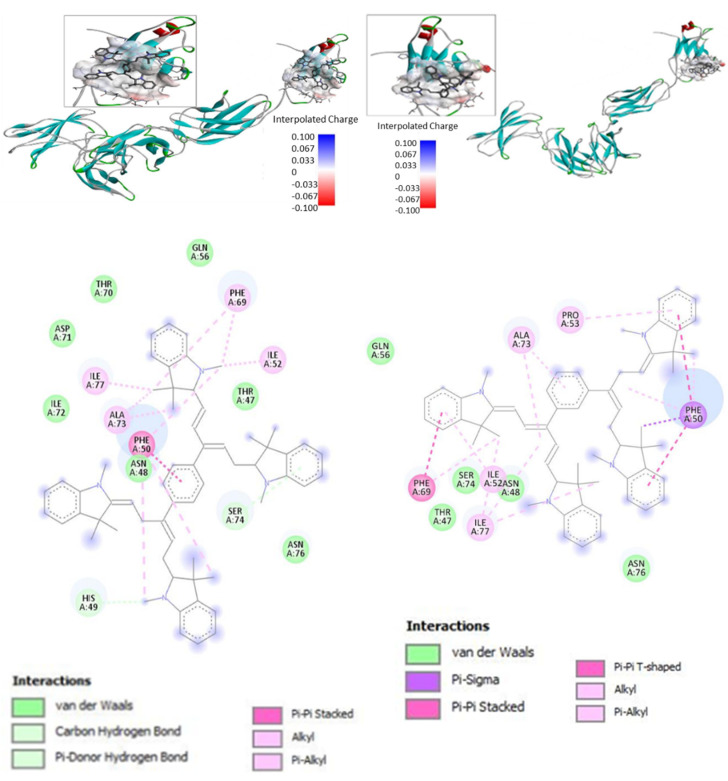
Docking of Bis-PMSs **5** and **6** to the IL-6 receptor (IL-6R) containing α and β subunits. Upper panels show a 3D model of IL-6R; the binding site (cavity) is localized in IL-6Rβ. Electron density in the binding site is shown as a color scale from blue (**low**) to red (**high**). Lower panels show the interaction of Bis-PMSs **5** and **6** with the amino acid residues of a homology model of IL-6R. Bis-PMS **5** and **6** docked to the IL-6R, −9.5 and, −8.1 kcal/mol, respectively.

**Figure 4 pharmaceutics-14-01712-f004:**
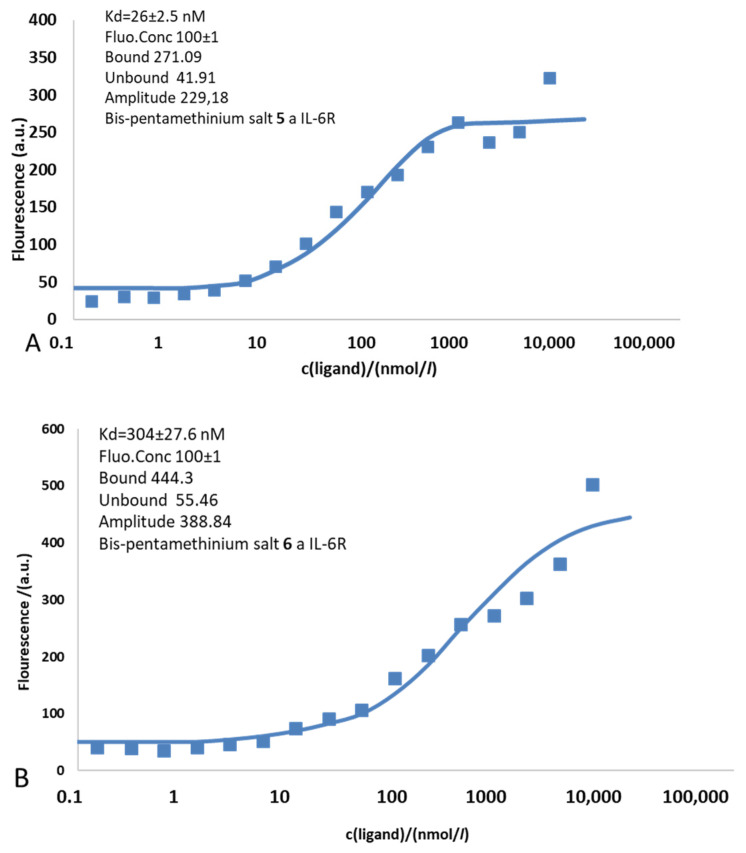
Binding of Bis PMS to the IL-6 receptor (IL-6R) containing α and β subunits evaluated by a microscale thermophoresis assay. The binding isotherm of Bis-PMSs **5** (**A**) and **6** (**B**) resulting from plotting the difference in normalized fluorescence against the concentration of the non-fluorescent binding partner (ligand–IL-6R).

**Figure 5 pharmaceutics-14-01712-f005:**
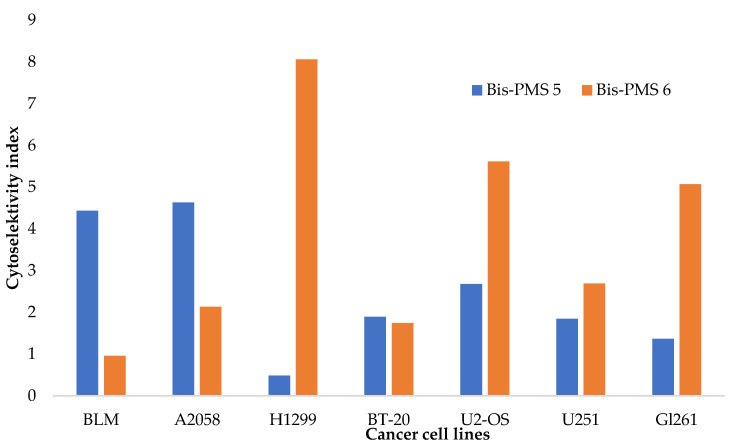
Cytoselectivity of Bis-PMSs **5** (blue) and **6** (orange) for cancer cell lines (BLM and A2058 human melanoma cells, H1299 human non-small cell lung cancer cells, human BT-20 breast cancer cells and U2-OS human osteosarcoma cells) against normal cells (HF P4 and BJ-hTERT human fibroblasts). For U251MG and Gl261 glioblastoma cells, the cytoselectivity was calculated against normal HF P4 fibroblast cells, which were analyzed using the same cytotoxicity assay.

**Figure 6 pharmaceutics-14-01712-f006:**
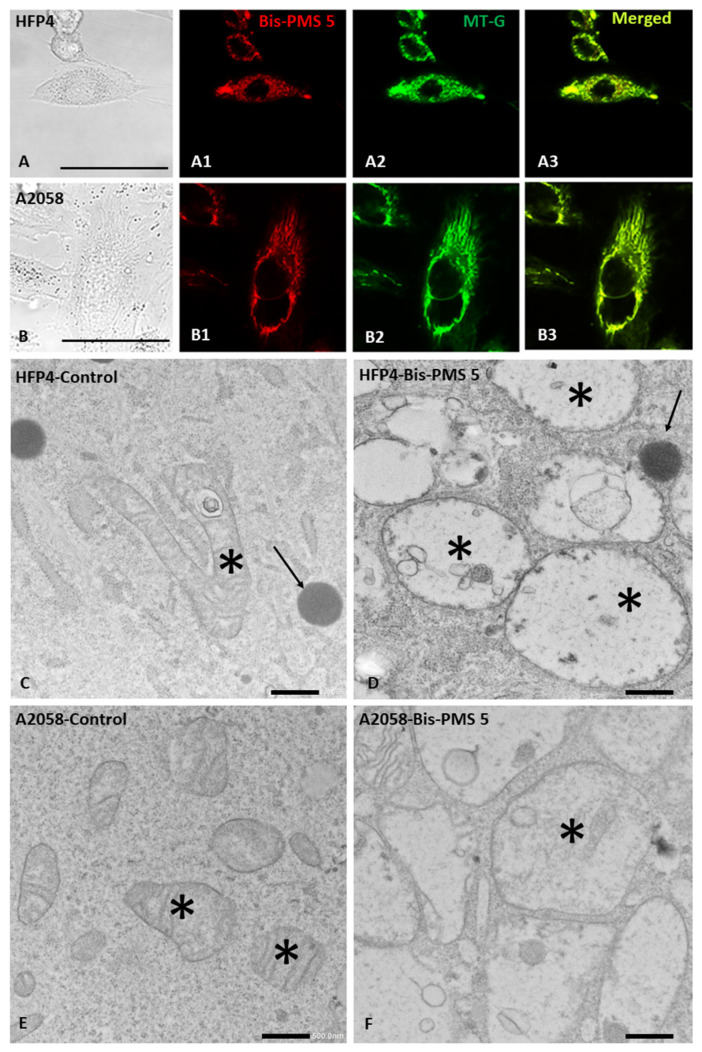
Influence of Bis-PMS **5** on normal HFP4 fibroblast (**A**,**C**,**D**) and A2058 melanoma cells (**B**,**E**,**F**). Subcellular localization of Bis-PMS 5 can be seen in the red channel (λex = 630 nm, λem = 650–670 nm, (**A1**,**B1**)) and corresponds to the position of mitochondria stained with Mito-Tracker (green channel: λex = 490 nm, λem = 510–516 nm; (**A2**,**B2**)). Colocalization is shown in (**A3**,**B3**) with a merged red and green signal. Electron microscopy (**C**–**F**) demonstrates the ultrastructure of normal fibroblasts (**C**), melanoma cells (**E**) and cells treated with Bis-PMS 5 (**D**,**F**). Exposure of both cell types to 5 µM of Bis-PMS 5 for 2 h induced swelling of mitochondria with cristolysis (**D**,**F**). Asterisks mark mitochondria and arrows mark lipid droplets. Bar is 100 μm (**A**,**B**) and 500 nm (**C**–**F**).

**Figure 7 pharmaceutics-14-01712-f007:**
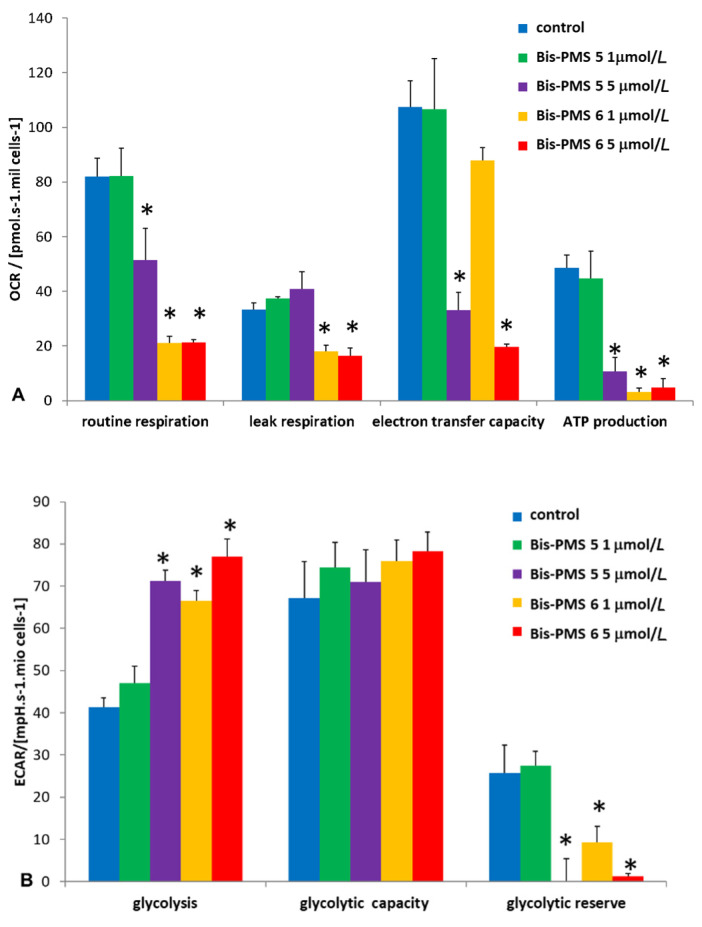
Influence of Bis-PMS on the energy metabolism in A2058 melanoma cells. **A** Seahorse Bioanalyzer was used to evaluate mitochondrial respiration (**A**) and glycolysis (**B**) in A2058 cells exposed to 1 and 5 μmol/L of Bis-PMSs **5** and **6** for 2 h. * *p* < 0.01 vs. control, *n* = 4, one-way ANOVA followed by a Tukey’s post hoc test. OCR = oxygen consumption rate, ECAR = extracellular acidification rate.

**Figure 8 pharmaceutics-14-01712-f008:**
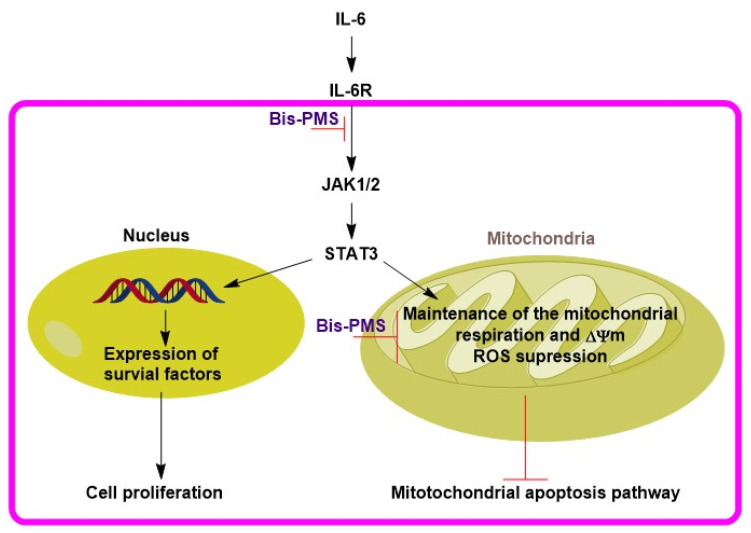
Proposed anticancer mechanism for Bis-PMSs **5** and **6**. Decreased activity of IL-6R via IL-6Rβ (gp130) targeting may cause lower levels of activated/phosphorylated STAT3. The decrease in STAT3 activity reduces expression of survival factors and mitochondrial functionality. As a result, cell proliferation is repressed and the mitochondrial apoptotic pathway is activated, respectively. Simultaneously, direct inhibition of mitochondrial metabolism by Bis-PMSs **5** and **6** leads to decreased cell growth and cell death.

**Table 1 pharmaceutics-14-01712-t001:** Cytotoxicity of Bis-PMSs **5** and **6** in normal and cancer cells. IC_50_ (µmol/L) of Bis-PMSs **5** and **6** was determined in HF-P4 and BJ-hTERT human fibroblasts, BLM and A2058 human melanoma cells, H1299 human non-small cell lung cancer cells, human BT-20 breast cancer cells, human U251MG and mouse Gl261 glioblastoma cells and U2-OS human osteosarcoma cells.

Cell Lines	Bis-PMS 5		Bis-PMS 6	
	Mean (µM)	SD	Mean (µM)	SD
**HFP4**	0.987	0.575	1.499	0.452
**BJ-hTERT**	1.660	0.038	1.218	0.442
**BLM**	0.300	0.117	1.419	0.486
**A2058**	0.287	0.109	0.636	0.359
**H1299**	2.755	0.063	0.169	0.031
**BT-20**	0.701	0.228	0.779	0.160
**U2-OS**	0.496	0.145	0.242	0.092
**U251**	0.719	0.219	0.505	0.020
**Gl261**	0.971	0.079	0.268	0.050

## Data Availability

Not applicable.

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
