# Peer review of "New-Generation Heterocyclic Bis-Pentamethinium Salts as Potential Cytostatic Drugs with Dual IL-6R and Mitochondria-Targeting Activity"

_pharmaceutics, 2022, doi:10.3390/pharmaceutics14081712_

Round 1
Reviewer 1 Report
Authors Talianova et al. in the paper entitled “New-generation heterocyclic bis-pentamethinium salts as potential cytostatic drugs with dual IL-6R and mitochondria-targeting activity’’ described the design, synthesis, and characteristics of two new bis-pentamethinium salts 5 and 6 (meta and para) bearing indole moieties.
1.Please, include the explanation for selected cancer cell line?
2. Results demonstrating the stability of tested compounds in cell culture medium for 48 h (at least) should be included.
3. MTT test does not detect cells proliferation.
4. Figure 5 should be modified to clearly demostrate cytototoxity on cancer cells vs non-cancer cells. Why the SD value for results obtained on non-cancer cells is so high? higher than the SD value for results obtained on cancer cells? Please clarify.
5. The results of TEM observation is not clearly and wrightly presented. Authors demontrate mitochondrial degradation in cells exposed to tested compounds. Please use control for each observation (at the same scale) and arrows to indicate ultrastructural changes (described in text or legend, respectively).
6. The discussion and conclusion of this manuscript is not clearly described. Please emphasize the role of tested compounds as potential anticancer drugs? Please take into account the limitation based on the obtained results.
Author Response
We would like to thank the reviewer for thorough reading and reviewing of our manuscript and especially for her/his remarks that helped us to significantly improve our manuscript. We have taken the reviewer’s advice and comments into account carefully point-by-point and made the following changes and corrections in the revised manuscript:
1.Please, include the explanation for selected cancer cell line?
We aimed to test the potential antitumor effect of the compounds against cancer cells representing various tumor types. Cell lines A2058 (CVCL_1059) and BLM (CVCL_7035) represent suitable models of aggressive malignant melanoma. Both cell lines were described in detail using various omics methods. Further, both cell lines are known to express IL-6 and IL-6R. The U251 and Gl261 cell lines represent frequently used and well characterized glioma cell models.
- Results demonstrating the stability of tested compounds in cell culture medium for 48 h (at least) should be included.
It is true, that the exposure of the cells in the MTT assay lasted 48 hours. Nevertheless, the mitochondrial uptake of the tested compound was found 30 minutes after their applications and their effect on the mitochondria was observed 2 hours after their application. Neverthelles, in protic solvents (e.g., water) polymethanium salts can be decomposed by solvolysis (hydrolysis in this case of water medium). In the used medium, both compounds, especially Bis-PMS 5 display significant spectral change over time (Fig. S4 and 5). But this effect was strongly repressed in the presence of MeOH (another protic solvent). Thus, the observed spectral change was not caused by their solvolysis, but most probably by their interaction with media components such as anionts, and/or aggregation. Based on the above, we can say that both Bis-PMS 5 and 6 are chemically stable, at least in the cell culture medium. In the revised version of the manuscript, this topic is discussed in the subchapter 3.2 Photophysical characteristic, lines 407-417.
- MTT test does not detect cells proliferation.
MTT is frequently used to assess cytotoxicity. In the manuscript, MTT was used to estimate the number of viable cells after exposure to the tested compounds. We agree that this number represents the result of cell proliferation and cell death, nevertheless the IC50 values obtained using the MTT test in our opinion reflect well the ability of the compounds to inhibit cell growth. “Cell proliferation” was removed from section 2.7. and 3.4.
- Figure 5 should be modified to clearly demostrate cytototoxity on cancer cells vs non-cancer cells. Why the SD value for results obtained on non-cancer cells is so high? higher than the SD value for results obtained on cancer cells? Please clarify.
Figure 5 was changed. In the current version of manuscript shoved cytoselectivity of tested compounds for cancer cells against normal fibroblast cell lines. Old version of figure 5 was moved into supplementary as figure S12. Mean value of IC50, including SD are now presented in the table 1 (requested by another reviewer).
Results from independent experiments are summarized, on average the SD values in cancer and normal cells are comparable (PMS 5 mean CV 30% in normal cells vs cancer cells CV 26%, PMS 6 mean CV 33% in normal cells vs cancer cells CV 27%). We added two sentences relating to the effect in normal vs cancer cells to section 3.5 (line 531 533), data on the cytoselectivity were included.
- The results of TEM observation is not clearly and wrightly presented. Authors demonstrate mitochondrial degradation in cells exposed to tested compounds. Please use control for each observation (at the same scale) and arrows to indicate ultrastructural changes (described in text or legend, respectively).
Figure 7 was amended to better illustrate the observed ultrastructural changes.
- The discussion and conclusion of this manuscript is not clearly described. Please emphasize the role of tested compounds as potential anticancer drugs? Please take into account the limitation based on the obtained results.
Usability and possible potential of the tested anticancer strategy and tested compounds, including possible limitations is discussed (line 598-657). To better illustrate the possible mechanism of anticancer effects, Figure 8 was also included.
Reviewer 2 Report
In this manuscript, the author design, synthesis, and characteristics of two new bis-pentamethinium salts 5 and 6 (meta and para) bearing indole moie-30 ties. They demonstrated that these two compounds can interact with IL-6R and accumulate in mitochondria. These two compounds inhibit mitochondrial respiration. Some points are needed to elucidate before publication.
1. The authors should investigate the effect of these two compounds on inhibiting IL-6 signaling pathway in cell-line by treating with IL-6.
2. In fig.5, the curves of cell viability of all cells by Bis-PMS 5 and 6 should be provided. The IC50 can be arranged and shown as a table.
3. A graphic model should be provided in manuscript for readers to inderstand.
Author Response
We would like to thank the reviewer for thorough reading and reviewing of our manuscript and especially for her/his remarks that helped us to significantly improve our manuscript. We have taken the reviewer’s advice and comments into account carefully point-by-point and made the following changes and corrections in the revised manuscript:
- The authors should investigate the effect of these two compounds on inhibiting IL-6 signalling pathway in cell-line by treating with IL-6.
This information was included in the manuscript. Effect of both compounds Bis-PMS 5 and 6 on the signalling was tested using a HEK reporter cell line. Application of both Bis-PMS 5 and 6 decreased secretion of alkaline phosphatase controlled by STAT3 in the HEK reporter cell line (Fig. S9). Nevertheless, the effect of tocilizumab was higher and it cannot be ruled out that part of the decrease in the secretion of alkaline phosphatase may partially be caused by cytotoxicity of Bis-PMS 5 and 6.
- In fig.5, the curves of cell viability of all cells by Bis-PMS 5 and 6 should be provided. The IC50 can be arranged and shown as a table.
The requested curves of cell viability were included in supplementary information (Fig. S10 and S11). The mean IC50s are now presented in the table 1 and IC50s represent curves of viability are shown in table S3. Original fig.5 was moved in the supplementary as figure S12. In the current version of manuscript figure showed cytoselectivity of tested compounds against normal fibroblast (requested by another reviewer).
- A graphic model should be provided in manuscript for readers to understand
To better illustrate the possible mechanism of anticancer effects, figure 8 was included and discussion was extended (lines 598-624).
Round 2
Reviewer 2 Report
The manuscript is well revised according to the reviewer's suggestions.